# Health Service Management and Patient Safety in Primary Care during the COVID-19 Pandemic in Kosovo

**DOI:** 10.3390/ijerph20043768

**Published:** 2023-02-20

**Authors:** Gazmend Bojaj, Bernard Tahirbegolli, Petrit Beqiri, Iliriana Alloqi Tahirbegolli, Esther Van Poel, Sara Willems, Nderim Rizanaj, Ilir Hoxha

**Affiliations:** 1Department of Health Institutions and Services Management, Heimerer College, 10000 Prishtina, Kosovo; 2Principal Family Medicine Center, 32000 Kline, Kosovo; 3National Sports Medicine Centre, 10000 Prishtina, Kosovo; 4Laboratory Technician Department, Heimerer College, 10000 Prishtina, Kosovo; 5Hematology Clinic, University Clinical Center of Kosovo, 10000 Prishtina, Kosovo; 6Department of Public Health and Primary Care, Faculty of Medicine and Health Sciences, Ghent University, 9000 Ghent, Belgium; 7Nursing Department, Heimerer College, 10000 Prishtina, Kosovo; 8The Dartmouth Institute for Health Policy and Clinical Practice, Geisel School of Medicine at Dartmouth, Lebanon, NH 03766, USA; 9Evidence Synthesis Group, 10000 Prishtina, Kosovo

**Keywords:** COVID-19, primary health care, PRICOV-19, quality of care, infection prevention and control, patient safety, family medicine, infectious diseases

## Abstract

Background: Several changes must be made to the services to ensure patient safety and enable delivering services in environments where the danger of infection of healthcare personnel and patients in primary care (PC) institutions is elevated, i.e., during the COVID-19 pandemic. Objective: This study aimed to examine patient safety and healthcare service management in PHC practices in Kosovo during the COVID-19 pandemic. Methods: In this cross-sectional study, data were collected using a self-reported questionnaire among 77 PHC practices. Results: Our main finding reveals a safer organization of PC practices and services since the COVID-19 pandemic compared to the previous period before the pandemic. The study also shows a collaboration between PC practices in the close neighborhood and more proper human resource management due to COVID-19 suspicion or infection. Over 80% of the participating PC practices felt the need to introduce changes to the structure of their practice. Regarding infection protection measures (IPC), our study found that health professionals’ practices of wearing a ring or bracelet and wearing nail polish improved during the COVID-19 pandemic compared to the pre-pandemic period. During the COVID-19 pandemic, PC practice health professionals had less time to routinely review guidelines or medical literature. Despite this, implementing triage protocols over the phone has yet to be applied at the intended level by PC practices in Kosovo. Conclusions: Primary care practices in Kosovo responded to the COVID-19 pandemic crisis by modifying how they organize their work, implementing procedures for infection control, and enhancing patient safety.

## 1. Introduction

Quality improvement in health facilities and the health system is essential to delivering appropriate healthcare [1,2]. This is especially important for primary care (PC) providers worldwide, who are the gateway to the healthcare system, and currently dealing with rising demands to match patients’ expectations for higher-quality medical care services and the quickening pace of scientific and technological advancement [1,2].

Globally, the COVID-19 pandemic has turned attention to PC and outpatient services [3]. In the newly imposed situation, the treatment of COVID-19 has taken up many ordinary healthcare resources [4]. As a result, hospitals’ non-COVID-19 primary and specialty care services for chronic and non-urgent care have mainly been scaled back or halted [4]. The COVID-19 pandemic has revealed the same weaknesses in PC and the healthcare system as in earlier pandemics [5]. PC, during the COVID-19 pandemic, has faced a decreased patient capacity and access to primary care, decreased quality of care, delays in the medical treatment of non-COVID patients, a rapid shift to alternative medical service delivery, and the requirements for adequate infection prevention and control (IPC) measures [6,7]. However, to perform their role as gatekeepers in authorizing access to hospital care, managing mild and moderate COVID-19 cases, performing diagnostic tests, and carrying out triage protocols to reduce the risk of overburdened hospitals, they needed to adapt to the new situation created by the pandemic era [3,8,9,10,11].

Furthermore, managing health systems during the COVID-19 pandemic was a key issue for ministries of health and the many governmental and nongovernmental organizations that deliver leadership in the healthcare field. This calls for inclusive leadership that collaborates with a wide range of stakeholders outside the public sector, from communities to researchers and academics, from doctors to civil society [12]. To adapt to the new era, reforms should have a focus that extends beyond “basic” service delivery and crosses established lines between the components that make up national health systems. If the mobilization around PC is informed by the lessons of past triumphs and failures, health officials may perform well in devising and implementing PC reforms tailored to individual country contexts and limits [13].

Since the first cases of COVID-19 [14], different incidence rates have been reported in Kosovo due to the different waves of the COVID-19 pandemic’s spread, and a peak in the disease’s mortality rate was recorded during July–August 2020 [15]. Furthermore, the COVID-19 pandemic had a psychological impact on healthcare practitioners, resulting in higher levels of depressive symptoms, COVID-19-related stress, tiredness, general anxiety, and decreased levels of proactive coping among healthcare workers [16,17,18,19]. On the other side, the infection of a significant number of PC workers with COVID-19, their quarantine, and a scarcity of personnel in their facilities have compounded the situation, necessitating the adaptation of health services for citizens [20].

PC in Kosovo is organized upon family medicine; care is delivered by a family medicine team, and family doctors are the main PC providers. Each municipality has a network of family medicine centers, including one main center and several affiliated PC centers. Furthermore, there are 1.94 visits per inhabitant per year and 2040 inhabitants per primary care physician. PC is supported by funding from the Kosovo government administered through municipalities. However, co-payments from patients are required for a standard set of laboratory tests and for each visit to a family medicine center. A formal referral from PC is required to consult with specific specialists in secondary care facilities [21,22].

Several changes must be made to the services to ensure patient safety and the continued delivery of services in a difficult environment where the danger of infection of healthcare personnel and patients in PC institutions is elevated during the COVID-19 pandemic. The PRICOV-19 initiative aims to discover which PC practice characteristics and healthcare system aspects are related to safe, effective, patient-centered, and equitable healthcare, as well as general practitioners’ mental health during COVID-19 pandemics in multiple countries [23]. This study focuses on patient safety and healthcare service management in PC practices in Kosovo during the COVID-19 pandemic.

## 2. Materials and Methods

### 2.1. Study Design and Setting

In the summer of 2020, an international consortium of researchers from 38 countries and more than 45 research institutions was formed under the coordination of Ghent University (Belgium) to set up the PRICOV-19 study. This multi-country cross-sectional study examined how PHC practices were organized during the COVID-19 pandemic to guarantee high-quality care, how the task roles changed and the pandemic impacted the well-being of care providers, and whether differences could be found between types of practices and or healthcare systems [23]. A coordinating center was set up in each country. In Kosovo, PRICOV-19 was coordinated by the Heimerer College’s Department of Management of Health Institutions and Services in Prishtina.

### 2.2. Measurement

Data were collected using an online self-reported questionnaire by the PC practices. The questionnaire was developed at Ghent University in multiple phases, including a pilot study among 159 PC practices in Flanders (Belgium). More details are described in the protocol [23]. The questionnaire consists of 53 items divided into the following main topics: infection prevention, patient flow for COVID-19 and non-COVID-19 care, dealing with new knowledge and protocols, communication with patients, collaboration, the well-being of the respondent, and characteristics of the respondent and the practice. The questionnaire was translated into the Albanian language following a standard procedure. Since the study comprised numerous countries, there were many types of PC structures and management. As a result, to best portray the reality and factual situation in each country in the research instrument, the researchers exchanged clarifying questions prior to the start of data collection and after it was completed. The Research Electronic Data Capture (REDCap) platform was used to host the questionnaire and securely store the participants’ answers [24].

### 2.3. Sampling and Recruitment

Between December 2020 and April 2021, data were collected from Kosovar PC practices. Through the use of email, we distributed the questionnaire to 105 PC practices randomly selected from all seven regions of Kosovo (Prishtina, Mitrovica, Peja, Prizren, Ferizaj, Gjilan, and Gjakova). The questionnaire in the electronic form was not required to answer all of the questions when submitted. As a result, some study participants did not answer all of the questions, resulting in missing data. This is also one of the reasons we decided to include in-paper completion of the questionnaire as an approach. PC practices that did not respond within a week were contacted by phone, and afterward, they were visited by a member of our research team, who provided the in-paper questionnaire.

The researcher returned after a week and collected the questionnaires. A different researcher electronically inputted the data from the collected questionnaires to reduce bias. Only 77 PC practices completed the questionnaire, generating a 73.3% response rate. For each PC practice, a questionnaire was filled out by a family doctor/general practitioner (GP), a GP trainee, or a staff member familiar with the practice’s organization. All participants provided written informed consent before enrolling in the study, and the participation was voluntary and anonymous.

### 2.4. Data Analysis

For categorical variables, frequencies and valid percentages were used to describe the variables.

The Chi-square test was used to compare the measurements before and during the pandemic. Statistical analysis was performed using SPSS software (version 21.0 SPSS Inc., Chicago, IL, USA). A *p*-value of <0.05 was considered statistically significant for all statistical tests.

### 2.5. Ethics Approval

The research was carried out following the Helsinki Declaration principles. The PRICOV-19 study protocol and Belgian data collection were authorized by the Research Ethics Committee at Ghent University Hospital (BC-07617).

## 3. Results

### 3.1. Sample Characteristics

Table 1 presents the study sample characteristics in Kosovo. In total, 42.8% (n = 33) of the PC practices were located in large cities. GP trainees completed 35% of the questionnaires. More than half of practices employed more than 15 health workers. One-fifth of the PC practices employed up to two GPs or GP trainees, and roughly one-third of PC practices provided services to populations larger than 10,000 people.

### 3.2. Patient Safety and Infection Prevention Measures

Table 2 reports the changes in patient safety and infection prevention measures at the Primary Healthcare Practices in Kosovo. In PC practices, the number of staff members that wear nail polish since the COVID-19 pandemic compared to the period before has decreased from 65% to 61% (*p* < 0.05). The number of practices that use a detailed protocol when cleaning the cleaning employees, each GP consultation room is equipped with hand sanitizer, and hand sanitizer is provided for patients at the door or waiting room of this practice, has been increased since the COVID-19 pandemic compared with the period before (from, 70% to 83%; 79% to 95%; and 69% to 95%, respectively) (*p* < 0.05) (Table 2).

### 3.3. Reviewing Guidelines or Reliable Scientific Evidence

Table 3 presents the availability of time for GP to review guidelines or go through relevant and reliable scientific literature. Compared to before the COVID-19 pandemic period, the percentage of PC practices that declare that there is enough protected time provided in the agenda(s) of GPs for reviewing guidelines or going through relevant and reliable scientific literature has increased (from 66% to 76%) (*p* < 0.05).

### 3.4. Infrastructure

Overall, 80% of the PC practices declared that since the COVID-19 pandemic, they had experienced any limitation related to the building or the infrastructure of the practice to provide high-quality and safe care. Additionally, 83.3% of PC practices declared that the COVID-19 pandemic led the practice to consider making adjustments in the future to the building or the infrastructure.

### 3.5. Availability of Administrative Documents

Figure 1 reports the availability of administrative documents at the Primary Healthcare Practice for suspected COVID-19 patients. If (suspected) COVID-19 patients need administrative documents (no prescriptions), these have been available for pickup in 28% of practices, and 9% have been sent to patients by regular email always (Figure 1). However, 19% of practices never sent these administrative documents to patients by postal mail or dropped them in the home letterbox of patients (Figure 1).

### 3.6. Support Services at the Primary Healthcare Practices

Figure 2 presents the results of support services at the Primary Healthcare Practices aimed at improving patient safety during the COVID-19 pandemic. We found that 28% of practices declared that they had sufficient time always between consultations for the disinfection of the consultation room. In addition, 18% of practices declare that if an incident about quality of care occurs in their practice, that is discussed at a(n) (online) team meeting (either with the whole team or only with the health professionals) always. However, 24% of practices declare that they sometimes use the protocol when answering patients’ phone calls and conducting telephonic triage.

Figure 3 presents the results of support services at the Primary Healthcare Practices aimed at improving patient health since the COVID-19 pandemic. We found that 74% of the practices agree that since the COVID-19 pandemic, staff members are more involved in giving information and recommendations to patients contacting the practice by phone. Next, 72% agree that since the COVID-19 pandemic, GPs or GP trainees are more involved in actively reaching out to patients that might postpone healthcare, and 66% state that since the onset of the COVID-19 pandemic, staff members are more involved in giving information or explaining what a caregiver has said to illiterate patients, patients with low health literacy or migrants (Figure 3).

### 3.7. Organization of Healthcare and Collaboration

Figure 4 reports the organization of healthcare and collaboration between PC practices to deliver healthcare services for patients during COVID-19. Overall, 41% of practices strongly agree that the COVID-19 pandemic has promoted cooperation with other PC practices in the neighborhood. Furthermore, 43% agree that if the staff members in practice are absent because of COVID-19 (infection or quarantine), the practice can count on the help of other PHC practices in the neighborhood. Moreover, 84% of the PC practices agree that if staff members in that practice are absent because of COVID-19 (infection or quarantine), the work can be distributed in such a way that the well-being of colleagues is not compromised.

## 4. Discussion

### 4.1. Summary of Results

Challenges in providing coordinated, comprehensive, continuous, and accessible care, along with rising pressure on GPs’ well-being, may have threatened GPs’ capacity to play a crucial role during the COVID-19 pandemic. As a result, the PRICOV19 project, a multi-country study that focuses on how GP practices deal with the issues of the COVID-19 pandemic, had substantial results [23].

Our main finding reveals a safer organization of PC practices and services since the COVID-19 pandemic compared to the previous period before the pandemic, as well as collaboration between PC practices in the close neighborhood and more proper human resource management due to COVID-19 suspicion or infection. Over 80% of the participating PC practices felt the need to introduce changes to the structure of their practice. In terms of infection protection measures (IPC), our study found that health professionals’ practices of wearing a ring or bracelet and wearing nail polish improved during the COVID-19 pandemic compared to the previous time. During the COVID-19 pandemic, PC practice health professionals had less time to review guidelines or medical literature routinely. Despite this, implementing triage protocols over the phone has yet to be applied at the intended level by PC practices in Kosovo.

### 4.2. Our Study in the Context

Although the improvement in not wearing nail polish by 4% from the study participants appears to be very small, the use of a detailed protocol when cleaning the cleaning employees increased by 13%, the GP consultation room equipped with hand sanitizer increased by 16%, and the availability of hand sanitizer for patients at the practice’s door or waiting room increased by 26%, since the onset of the pandemic. These results of our study, which emphasize safer services and the prevention of infection spread in PC practices, are consistent with recent publications and the existing literature on how health professionals‘ practices improved during the COVID-19 pandemic [25,26,27,28]. Similarly, all countries taking part in the PRICOV study recorded improvements in IPC measures in PC practices since the COVID-19 pandemic [29]. On the other hand, it is evident that the pandemic has reduced the number of primary care health services and supplies of medical preparations, discontinuance of “non-essential” services, and staff shortages due to fear of infection and lack of PPE, reducing nurse–patient relationships, replacing face-to-face services with online consultations, and telephone triage [11,20,30,31,32,33].

Although, healthcare professionals cite several factors that affect their capacity and willingness to adhere to IPC guidelines, including how the guidelines are communicated, their desire to provide quality patient care, manager support, workplace culture, training, physical space, access to and trust in personal protective equipment when managing respiratory infectious diseases [34].

In their study, Lau et al. found that while GPs in Singapore are concerned about infection prevention, access to information on COVID-19, and the well-being of their colleagues and family, they accept the risks and the need to care for COVID-19 patients [35]. Lim et al. provide a scoping review of potential solutions to reduce the impact of COVID-19 on PC and emphasize that PC services take a unified, flexible, and evidence-based strategy to address the problems posed by the COVID-19 pandemic [6]. Further, in their study, Patel et al. demonstrated that training and raising the awareness of the medical staff regarding the mechanisms of preventing the spread of infection in emergency conditions such as the COVID-19 pandemic has positive effects on improving IPC outcomes and decreasing infections among the medical staff in PC [36].

In contrast to our findings, other countries that participated in the PRICOV19 survey, such as Denmark, Switzerland, and Latvia, on approximately 50% of PC practices, did not consider future building or infrastructure changes [37]. However, large differences were noted between countries in the experienced limitations to the building or other practice infrastructure to provide high-quality and safe care during the COVID-19 pandemic and considering making adjustments to the building or infrastructure, which may be related to the organization of the PC and the country’s economic development [37].

Although PC services in Kosovo are widely available without prior appointments, and PC centers are open at least five days a week for at least seven hours every day. All main family medical centers and a few additional PC practices in larger municipalities offer 24 h services. Before the COVID-19 pandemic, in PC, there were insufficient proactive and preventive people-centered approaches that take vulnerable population groups’ health needs into account [21].

According to our findings, since the COVID-19 pandemic, staff members of PC practices have been more involved and proactive in patient care, as well as in distributing work in the case that a staff member in practice is missing owing to COVID-19 infection/quarantine. This is in line with the findings of the Groenewegen et al. study, indicating that PC practices also changed the tasks of healthcare personnel in response to the needs for care during the COVID-19 pandemic [38]. Many PRICOV study participants agreed that their responsibilities had increased and that GPs and GP trainees were more involved in proactively reaching out to patients who might delay seeking healthcare [38]. However, pandemics are high-stakes situations, and PC workers should frequently manage difficult and unacknowledged burdens of responsibility. As a result, there is a need for improved collaboration and communication between governments and PC [5].

### 4.3. Study Strengths and Limitations

It is also important to recognize some limitations of the study. First, our survey’s use of a self-reported questionnaire, which is biased by nature, is one of its limitations. Second, online and in-paper methods were used to acquire the data. Third, the data collection encompassed several pandemic phases because it was conducted over 18 weeks. The study’s strength, however, is that the questionnaire used was developed and validated over multiple stages, including a pilot study. Second, compared to most other countries, the Kosovar PRICOV19 data has a high response rate.

## 5. Conclusions

PC practices in Kosovo responded to the COVID-19 pandemic crisis by making several modifications to the way they organize their work, implementing procedures for infection control and enhancing patient safety. Since the COVID-19 pandemic, a substantial proportion of PC practices have experienced limitations connected to the practice’s facility or infrastructure in providing high-quality and safe care and have felt the need to change the structure of their practice. Thus, the government should offer systematic support for the development of practice infrastructure to deliver high-quality, safe primary care in the event of future emergencies similar to the COVID-19 pandemic. The preparedness of the health system and health organizations for response in unusual circumstances, including pandemics, is essential for providing optimal health care for citizens.

## Figures and Tables

**Figure 1 ijerph-20-03768-f001:**
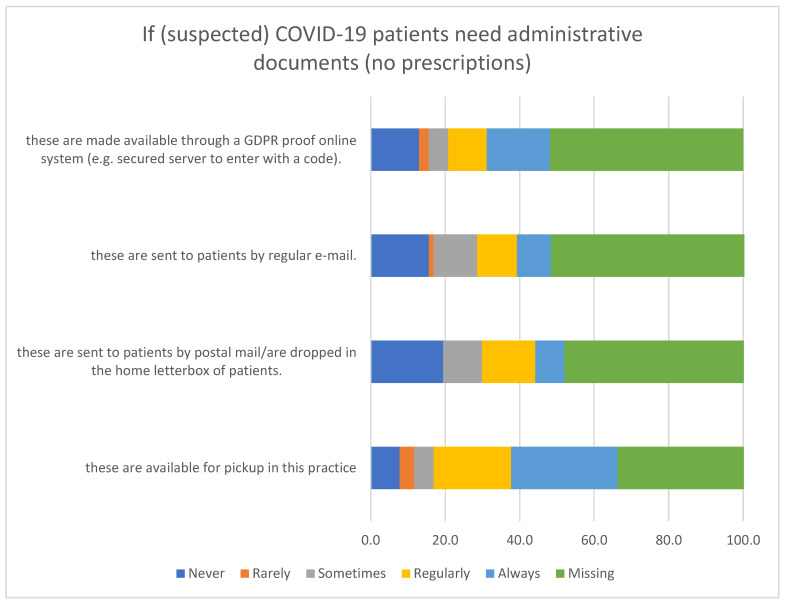
Availability of administrative documents at the Primary Healthcare Practice for suspected COVID-19 patients.

**Figure 2 ijerph-20-03768-f002:**
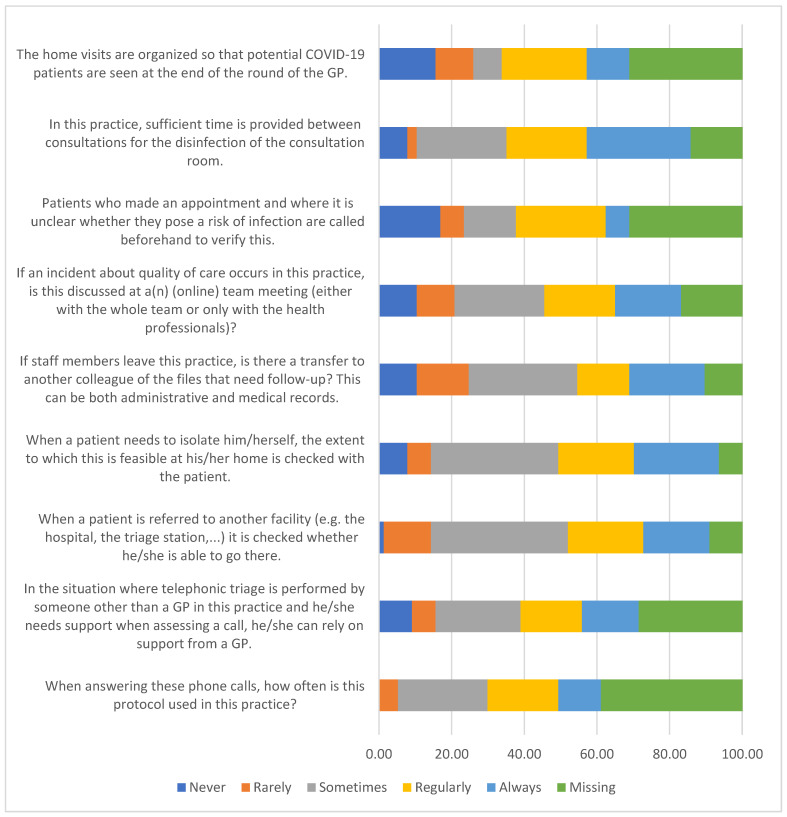
Support services at the Primary Healthcare Practices to improve patient safety during the COVID-19 pandemic.

**Figure 3 ijerph-20-03768-f003:**
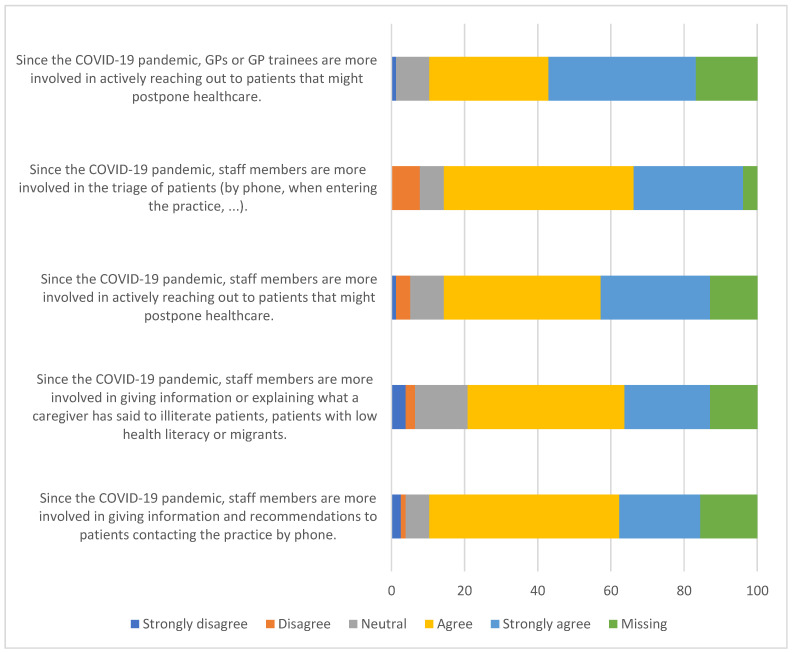
Support services at the Primary Healthcare Practices to improve patient health since the COVID-19 pandemic. GP = general practitioner.

**Figure 4 ijerph-20-03768-f004:**
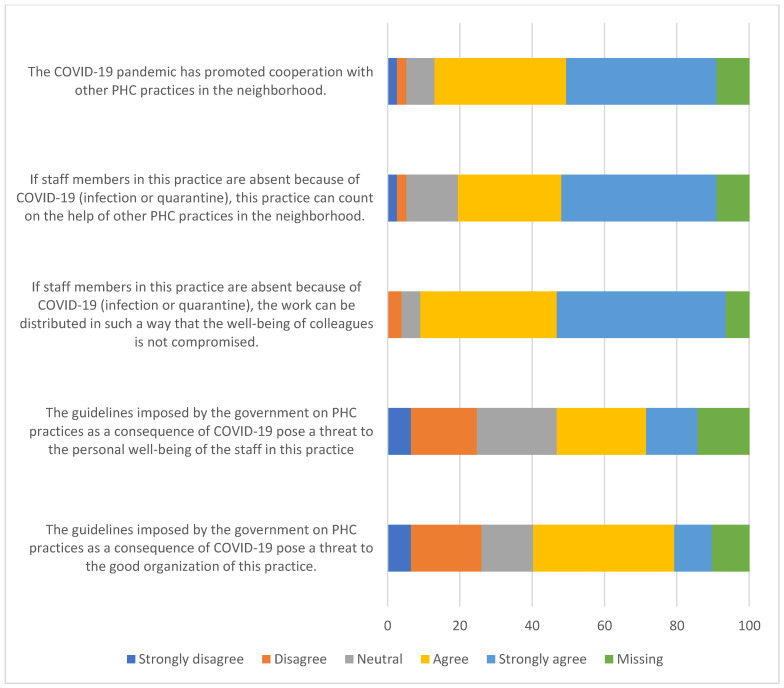
Organization of healthcare and collaboration between PHC practices to deliver healthcare services for patients during COVID-19.

**Table 1 ijerph-20-03768-t001:** Characteristics of Kosovar respondents of the PRICOV-19 questionnaire.

	N (%)
Position in the practice	
GP	39 (50.65)
GP trainee	27 (35.06)
Other	5 (6.49)
Missing	6 (7.79)
Location of the practice	
Big (inner) city	33 (42.86)
Suburbs	7 (9.09)
(Small) town	21 (27.27)
Mixed urban-rural	10 (12.99)
Rural	6 (7.79)
Number of GPs or GP trainees in the practice	
1–2	15 (19.48)
3–4	13 (16.88)
5–6	13 (16.88)
>6	31 (40.26)
Missing	5 (6.49)
Number of paid staff	
1–5	3 (3.90)
6–10	10 (12.99)
11–15	4 (5.19)
>15	42 (54.55)
Missing	18 (23.38)
Number of outpatients per practice	
0–5000	12 (15.58)
5001–10,000	9 (11.69)
>10,000	24 (31.17)
Missing	32 (41.56)

**Table 2 ijerph-20-03768-t002:** Changes on the patient safety and infection prevention measures at the Primary Healthcare Practices in Kosovo.

Since the COVID-19 Pandemic						
**Before the COVID-19 Pandemic.**			One or more staff members wear nail polish.			
			Never	Sometimes	Always	
	One or more staff members wear nail polish	Never	18 (25.0)	4 (5.6)	0 (0.0)	*x*^2^ = 34.069 *p* < 0.0001
		Sometimes	7 (9.7)	28 (38.9)	6 (8.3)	
		Always	0 (0.0)	6 (8.3)	3 (4.2)	
			One or more staff members wear a ring or bracelet.			
	One or more staff members wear a ring or bracelet.	Never	14 (19.2)	5 (6.8)	2 (2.7)	*x*^2^ = 38.721 *p* < 0.0001
		Sometimes	6 (8.2)	22 (30.1)	4 (5.5)	
		Always	1 (1.4)	6 (8.2)	13 (17.8)	
			When cleaning, the cleaning employees use a detailed protocol (e.g. what to clean, frequency, method).			
	When cleaning, the cleaning employees use a detailed protocol	Never	10 (13.9)	5 (6.9)	8 (11.1)	*x*^2^ = 28.262 *p* < 0.0001
		Sometimes	3 (4.2)	17 (23.6)	6 (8.3)	
		Always	0 (0.0)	6 (8.3)	17 (23.6)	
			Each GP consultation room is equipped with hand sanitizer.			
	Each GP consultation room is equipped with hand sanitizer	Never	4 (5.4)	3 (4.1)	9 (12.2)	*x*^2^ = 16.741 *p* = 0.002
		Sometimes	0 (0.0)	3 (4.1)	12 (16.2)	
		Always	0 (0.0)	5 (6.8)	38 (51.4)	
			Hand sanitizer is provided for home visits.			
	Hand sanitizer is provided for home visits	Never	4 (5.6)	8 (11.1)	3 (4.2)	x^2^ = 32.825 *p* < 0.0001
		Sometimes	0 (0.0)	4 (5.6)	19 (26.4)	
		Always	0 (0.0)	3 (4.2)	31 (43.1)	
			Hand sanitizer is provided for patients at the door or waiting room of this practice.			
	Hand sanitizer is provided for patients at the door or waiting room of this practice.	Never	4 (5.6)	6 (8.5)	14 (19.7)	*x*^2^ = 18.800 *p =* 0.001
		Sometimes	0 (0.0)	3 (4.2)	6 (8.5)	
		Always	0 (0.0)	1 (1.4)	37 (52.1)	
			A separate bag is provided for home visits to patients with suspected infection.			
	A separate bag is provided for home visits to patients with suspected infection.	Never	11 (15.7)	5 (7.1)	8 (11.4)	*x*^2^ = 40.398 *p* < 0.0001
		Sometimes	0 (0.0)	9 (12.9)	6 (8.6)	
		Always	0 (0.0)	3 (4.3)	28 (40.0)	

Data are given as n (valid percentage); GP = general practitioner.

**Table 3 ijerph-20-03768-t003:** There is enough protected time provided in the agenda(s) of GPs for reviewing guidelines or going through relevant and reliable scientific literature.

		Since The COVID-19 Pandemic				
		Disagree	Neutral	Agree	Strongly Agree	
**Before The COVID-19 Pandemic**	**Disagree**	3 (4.4)	0 (0.0)	0 (0.0)	0 (0.0)	*x*^2^ = 37.160 *p* < 0.0001
	**Neutral**	1 (1.5)	8 (11.8)	7 (10.3)	4 (5.9)	
	**Agree**	0 (0.0)	4 (5.9)	16 (23.5)	11 (16.2)	
	**Strongly Agree**	0 (0.0)	0 (0.0)	5 (7.4)	9 (13.2)	

Data are given as n (valid percentage); GP = general practitioner.

## Data Availability

The anonymized data are held at Ghent University and are available to participating partners for further analysis upon signing an appropriate usage agreement.

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
