# Peer review of "Health Service Management and Patient Safety in Primary Care during the COVID-19 Pandemic in Kosovo"

_ijerph, 2023, doi:10.3390/ijerph20043768_

Round 1

Reviewer 1 Report

Thank you for this interesting paper providing some useful insights into changes in primary care in Kosovo because of COVID-19.

The paper is generally well written and structured, though I have some recommendations that I hope will improve it overall.

Here are my comments on each section.

INTRODUCTION

This section provides a helpful background. Sources are used appropriately, and I can see that you are seeking to justify your study, which is part of a larger multi-country study.

I think here it would be useful to hear more about primary health care in Kosovo – for example, what is its key role? Is it a key gatekeeper to secondary care services as elsewhere? Do patients generally access easily and turn up for appointments? Is payment involved? This will differ between countries so knowing about the situation in Kosovo would be useful.

MATERIALS AND METHODS

This section provides a detailed overview of your methodology and how you gathered and analysed the data. I note that you had ethical approval for the study, and you obtained consent from participants.

A minor typo on line 112 – it should be ‘data were collected’.

RESULTS

These provide some interesting insights into changes brought about because of the COVID-19 pandemic. I note in Table 1 quite a few ‘missing’ statistics – is there a reason for this? This is also evident in other findings, and I’d recommend saying more in the text as to why this should be.

Tables 2 and 3 are interesting, though as these only describe the situation since COVID-19, it would be useful to have ‘before and after’ here (rather than just a couple of comments in the text). Are these data available?

Figures 1 and 2 (as above) have a lot of ‘missing’ – are these ‘don’t knows’ or is the data absent?

The reduction in carers not using nail polish since COVID-19 is very small, at 4% (according to the text – it’s not possible to see this in the Tables/Figures). Why was this the case? Can it be addressed in the discussion?

DISCUSSION

You draw on your findings and link this with what is currently known. You make some helpful points, but when you state that the organisation is ‘safer’ this is difficult to confirm as some of the data presented doesn’t have all the information (pre- and post-COVID-19). There have clearly been some improvements, but more information would be helpful for the readers to agree with that statement.

I would also recommend expanding the focus on Kosovo here (section 4.1) and connect this back to the specific role and status of primary care in Kosovo.

I note that you include strengths and limitations, which are appropriate.

CONCLUSION

This is useful but brief. A mention of the need for ‘pandemic preparedness’ would strengthen this section significantly – this is after all what you are implicitly recommending. There are multiple sources and citations that could be useful here.

REVIEWER RECOMMENDATIONS

1.     Discuss a little more in the introduction about the status and role of primary care in Kosovo in the introduction.

2.     Correct minor typo (data ‘was’ to data ‘were’).

3.     In the results section, provide more clarity about how data post-COVID-19 are different from pre-COVID-19. You could rework the tables?

4.     Add a note as to why there are ‘missing’ data.

5.     Broaden reference to Kosovo in the discussion section.

6.     Include a (short but important) note in the conclusion about pandemic preparedness. This will fit with your recommendation.

7.     Consider including in the discussion the one variable that changed only a little pre-/post-COVID-19 - the use of nail polish. Why was this so?

Author Response

Thank you for your comments.

We did our best to respond one by one to all the raised issues in the review.
Please find enclosed the response letter.

Reviewer 2 Report

Your manuscript is well written and provides generally solid data, it is clear and concise. While the international study you refer to and have taken part in is significant, the approached subject needs to be discussed a little bit more in detail in the discussions section. Since you approach such a broad subject as health service management during the COVID 19 pandemic, the authors should also include data regarding the negative side effects of the COVID 19 pandemic on the healthcare system.

Your conclusions focus rather strictly on the safer organization of PC practices and services since the pandemic began. Your positive outlook is quite admirable although there are several studies that have also observed negative outcomes of the COVID 19 pandemic regarding patient care as well as healthcare specialist mental health, taking this into consideration, this aspect deserves at least mentioning in your discussion section. With the purpose of aiding the authors, the following studies can be consulted

 Impact of COVID-19 pandemic on utilisation of healthcare services: a systematic review By Moynihan et al, DOI: 10.1136/bmjopen-2020-045343; The Impact of the COVID-19 Pandemic on Nursing Care: A Cross-Sectional Survey-Based Study by Clari et al doi: 10.3390/jpm11100945; Impact of COVID-19 Pandemic on the Implantation of Intra-Cardiac Devices in Diabetic and Non-Diabetic Patients in the Western of Romania by Pescariu et al DOI: 10.3390/medicina57050441; The Effects of the Health System Response to the COVID-19 Pandemic on Chronic Disease Management: A Narrative Review by Kendzerska et al. doi: 10.2147/RMHP.S293471.

A short paragraph in which the authors also discuss the broader negative aspects of the pandemic on patient management would be a welcome addition to the manuscript.

Also minor corrections are necessary, for example 3.2 Our study instead of Out study (line 231)

Author Response

(The authors gave the same response as above.)

Round 2

Reviewer 1 Report

Thank you for this revised version of your paper. I have no further comments, as I can see that you have addressed my recommendations. Thanks for your work on this.